# Hematological and Serum Biochemical Reference Intervals for Alphaxalone Sedated Common Marmosets (*Callithrix jacchus*)

**DOI:** 10.3390/ani14050790

**Published:** 2024-03-03

**Authors:** Merel Wegman, Jaco Bakker, Remco A. Nederlof, Edmond J. Remarque, Jan A. M. Langermans

**Affiliations:** 1Animal Science Department, Biomedical Primate Research Centre, Lange Kleiweg 161, 2288 GJ Rijswijk, The Netherlands; bakker@bprc.nl (J.B.); langermans@bprc.nl (J.A.M.L.); 2Department Population Health Sciences, Animals in Science and Society, Faculty of Veterinary Medicine, Utrecht University, 3584 CM Utrecht, The Netherlands; remnederlof@hotmail.com; 3Department of Virology, Biomedical Primate Research Centre, Lange Kleiweg 161, 2288 GJ Rijswijk, The Netherlands; remarque@bprc.nl

**Keywords:** aging, alphaxalone, biochemistry, *Callithrix jacchus*, contraception, hematology, housing condition, marmoset, pregnancy, reference intervals

## Abstract

**Simple Summary:**

In the present study, hematological and serum biochemical reference intervals were established for common marmosets sedated with alphaxalone through all life stages, including pregnancy and contraception. Clinically relevant changes were established for multiple parameters when comparing data based on age, sex, housing condition, contraceptive use, and pregnancy. The availability of updated reference intervals, calculated using a large sample size and using the appropriate statistical methods, is an essential part of providing optimum care and support to marmosets used in biomedical research.

**Abstract:**

Marmosets are routinely used in biomedical research, therefore there is an increasing need for updated reference intervals calculated using a large sample size, correct statistics, and considering different variables. Hematological and biochemical values from 472 healthy common marmosets sedated with alphaxalone were collected over a ten-year period (2013–2023). The variables assumed to have influenced the blood-based parameters were compared, i.e., sex, age, housing condition, pregnancy, and contraceptive use. Reference intervals were calculated based on observed percentiles without parametric assumptions, and with parametric assumptions following Box–Cox transformation. Juvenile marmosets showed increased ALP, phosphate, WBC, lymphocyte count, and basophil count and decreased levels of GGT and Fe compared to adults. Marmosets housed strictly indoors showed increased ALT and GGT levels and decreased levels of total bilirubin and neutrophil count compared to marmosets housed with outdoor access. Pregnant marmosets showed increased ALP, total bilirubin, neutrophil count, monocyte count, and basophil count, and decreased levels of AST, ALT, cholesterol, Fe, and lymphocyte count compared to non-pregnant marmosets. Etonogestrel contracepted marmosets showed decreased P-LCR compared to females who were not contracepted. Updated reference intervals will aid researchers and veterinarians in identifying physiological and pathological changes, as well as improve the reproducibility of research in this species.

## 1. Introduction

Common marmosets (*Callithrix jacchus*) are a species of nonhuman primates (NHP) that are routinely used in biomedical research around the world [1]. Knowledge of marmoset biology, anatomy, physiology, and behavior is necessary to optimize the care for this species. Hematological (HER) and serum biochemical (CER) values are considered important components of health monitoring, improving the utility of this species as a model in biomedical research.

A comparative analysis of the current literature concerning HER and CER reference intervals for marmosets showed that these intervals were either outdated, calculated using a relatively low number of animals, or calculated using parametric assumptions that were not warranted [2,3,4,5,6,7,8].

Therefore, the HER and CER values collected from a large cohort of healthy common marmosets were used to calculate these reference intervals. Because many HER and CER parameters are not normally distributed, the data used in this study are calculated with nonparametric methods as observed percentiles and with parametric methods following the Box–Cox transformation to obtain a normal distribution. It was hypothesized that the HER and CER values are influenced by age, sex, housing condition (indoor versus outdoor), pregnancy, and contraceptive use. This study presents veterinarians and biomedical researchers with an updated set of reference intervals of common marmosets while applying the appropriate statistical methodology.

## 2. Materials and Methods

The data used to create the reference intervals were acquired retrospectively from the electronic health records of the Biomedical Primate Research Centre (BPRC, Rijswijk, The Netherlands). Blood was collected from marmosets during routine health surveillance. Therefore, ethical approval was not necessary for this study. All procedures and husbandry were in accordance with Dutch law and international ethical and scientific standards and guidelines (EU Directive 63/2010, Weatherall report, Guide for the Care and Use of Laboratory Animals, and standards outlined by AAALAC).

Every year, as part of the routine annual health surveillance, each marmoset underwent a complete physical examination. This included abdominal palpation for pregnancy assessment and the determination of the body condition score (BCS) [9,10]. To monitor the tuberculosis status all marmosets were given a tuberculin skin test. Feces were sampled in order to screen for various pathogens. The health program did not include anti-parasitic treatment. These examinations were performed annually for each marmoset. To prevent outbreaks of *Yersinia enterocolitica* and *Y*. *pseudotuberculosis* in the colony, all marmosets received a Pseudovac^®^ vaccination biannually (Utrecht University, Department of Veterinary Pathology, Zoo and Exotic Animals Section, Utrecht, The Netherlands).

Twice daily, animal caretakers observed all marmosets and checked for injuries, illness, and fecal consistency, which was recorded daily in electronic health records for each animal. Parturitions, abortions, and stillbirths were also recorded in the electronic health records. If the caretaker took note of any abnormalities, they were reported to a veterinarian. The veterinarians regularly checked all animals and when an animal showed symptoms of disease, a veterinary record was created documenting the symptoms, treatment, and progression of the disease. These records were used to determine whether an animal could be included in the determination of reference intervals. Exclusion criteria included the presence of a veterinary record created within 6 weeks of blood collection to prevent inclusion of (subclinical) disease, a BCS lesser than 1.5 or greater than 4.5, or significant weight loss. Experimental animals or animals with a history of experimental use were also excluded.

For colony management several female marmosets received a 23 mg etonogestrel contraceptive implant (Implanon NXT^®^, N.V. Organon, Oss, The Netherlands) subcutaneously between the shoulder blades [11].

### 2.1. Animals

The marmoset colony at the BPRC was originally founded in 1975 and initially was made up of marmosets obtained from various suppliers. The outbred characteristics of the colony were maintained by the regular introduction of genetically non-related animals. All study animal pedigrees and birth dates were known at the time of blood collection. The colony consisted of approximately 15 breeding groups alongside a group of pair-housed marmosets to be used in experiments. Together making up approximately 150 marmosets on average with ages ranging from neonatal to geriatric animals.

Data used in this study were collected from 1 October 2013 to 24 April 2023. Only animals aged over five months and weighing > 200 g were sedated, resulting in an age range of 0.5–10 years. The dataset consisted of a total of 472 animals and included data on body weight, BCS, age, sex, pregnancy, contraception, and housing conditions. Table 1 shows the sample size with the number of unique animals split by age and sex. Over a period of ten years, with one physical examination per animal per year, a total of 1310 blood samples were included in the dataset (Table 2).

### 2.2. Housing and Husbandry

The breeding colony was housed in enclosures with outdoor access. In these enclosures, breeding pairs were housed with multiple generations of their offspring. The offspring remained housed in the family group until either parent rejected them or until they were selected for experimental use. The minimum age for marmosets to be selected for experimental use at the BPRC is 1.5 years. Experimental animals were relocated to strictly indoor housing and housed socially in pairs. Therefore, all animals that are housed in indoor housing used to be housed with outdoor access. The details of both housing conditions are specified below.

#### 2.2.1. Outdoor Access

The enclosures consisted of an indoor and outdoor compartment measuring 300 × 200 × 300 cm. The indoor compartment was heated and maintained between 22 and 25 °C. The outdoor compartment was not heated. The animals had unrestricted access to both compartments. The indoor compartment included bedding material provided as floor covering. The outdoor compartment floor covering consisted of sand. Environmental enrichment in the cage was provided using various materials including branches, ropes, nets, garden and fire hoses, and wooden runways. Light was provided with a 12 h light/dark cycle using full spectrum fluorescent bulbs in addition to natural light though the windows in the compartments. The ventilation rate was eight air changes per hour and the relative humidity was kept between 50% and 60%. Diet consisted of commercial primate pellets for New World monkeys (Ssniff, Soest, Germany), supplemented with limited portions of fruit, vegetables, Arabic gum, live mealworms, and homemade porridge containing oats and Ssniff^®^ marmoset powder as main ingredients with added vitamin supplements. Tap water was provided ad libitum using automatic water dispensers.

#### 2.2.2. Strictly Indoor Housing but with a History of Outdoor Access

Experimental animals were housed in pairs, indoors, in cages measuring 150 × 75 × 185 cm, with bedding material provided as floor covering. The marmosets’ environmental enrichment was provided by ropes, nets, and wooden runways. Light was provided with a 12 h light/dark cycle using full spectrum fluorescent bulbs in addition to natural light though the windows in the room. Room temperature was maintained between 23 and 27 °C. Diet consisted of Ssniff^®^ primate pellets for New World monkeys (Ssniff, Soest, Germany), supplemented with limited portions of fruit, vegetables, Arabic gum, live mealworms, and homemade porridge containing oats and Ssniff^®^ marmoset powder as main ingredients with added vitamin supplements. Additional enrichment was given regularly according to an enrichment protocol and consisted of both food and non-food enrichment items. Tap water was provided ad libitum using water bottles.

### 2.3. Sedation, Blood Collection and Analysis

Blood samples were collected in the morning and processed the same day at the BPRC. Prior to sedation the animals were fasted by withholding food for 16 h. Water was never restricted. The animals were trained weekly using positive reinforcement training to voluntarily enter a balcony box attached to the cage. Subsequently, the animals were physically restrained with leather gloves and chemically immobilized by the intramuscular (IM) administration of 12 mg/kg alphaxalone (Alfaxan Multidose 10 mg/mL, Jurox Limited, London, UK) [12,13]. Alphaxalone is a GABA receptor agonist, resulting in muscle relaxation and sedation. The sedative agent is known to cause dose-dependent respiratory depression, but at the dosage used this effect is limited, and intubation is not required [13]. The sedative was administered by a caretaker into the musculus quadriceps femoris using a 26-gauge needle, while another caretaker manually restrained the animal. Following induction, the body weight was recorded. The skin in the groin was shaved and sterilized with a 70% alcohol solution. Blood was collected from the vena femoralis in 1 mL EDTA and 2 mL serum tubes using a 20-gauge Vacuette® needle and hub (Greiner Bio-One GmbH, Kremsmünster, Austria). 

Hematology samples were collected using K3-EDTA blood collection tubes (Greiner Bio-One GmbH, Kremsmünster, Austria) and mixed by inversion. Samples with clots were excluded from this study. HER values were measured using a Sysmex XT2000iV (Sysmex BV, Etten-Leur, The Netherlands). The measured parameters included a red blood cell count (RBC, 10^12^/L), hemoglobin (HGB, mmol/L), hematocrit (HCT, L/L), mean corpuscular volume (MCV, fL), mean corpuscular hemoglobin (MCH, amol), mean corpuscular hemoglobin concentration (MCHC, mmol/L), platelet count (PLT, 10^9^/L), red blood cell distribution width—standard deviation (RDW-SD, fL), red blood cell distribution coefficient of variation (RDW-CV, fL), platelet distribution width (PDW, fL), mean platelet volume (MPV, fL), platelet larger cell ratio (P-LCR,%), plateletcrit (PCT,%), white blood cell count (WBC, 10^9^/L), neutrophil count (Neut, 10^9^/L), lymphocyte count (Lymph, 10^9^/L), monocyte count (Mono, 10^9^/L), eosinophil count (Eo, 10^9^/L), and basophil count (Baso, 10^9^/L).

For serum biochemistry values, samples were collected in Vacuette® 2.5 mL CAT Serum Separator Clot Activator tubes (Greiner Bio-One GmbH, Kremsmünster, Austria). To obtain serum, blood was allowed to clot at room temperature for at least one hour before centrifugation at 3000 RPM for ten minutes. Hemolytic and lipemic serum samples were excluded from the study. CER values were measured using a Cobas Integra 400 plus analyzer (F. Hoffmann-La Roche Ltd., Basel, Switzerland). The measured parameters included albumin (ALB, g/L), total protein (TP, g/L), alkaline phosphatase (ALP, U/L), alanine aminotransferase (ALT, U/L), aspartate aminotransferase (AST, U/L), lactate dehydrogenase (LDH, U/L), gamma-glutamyltransferase (GGT, U/L), total bilirubin (TBIL, μmol/L), cholesterol (Chol, mmol/L), chloride (Cl, mmol/L), bicarbonate (BIC, mmol/L), iron (Fe, μmol/L), potassium (K, mmol/L), sodium (Na, mmol/L), phosphate (Phos, mmol/L), calcium (Ca, mmol/L), glucose (Glu mmol/L), urea (URE, mmol/L), and creatinine (Cre, μmol/L).

### 2.4. Investigated Variables

The investigated variables included sex, age, housing condition, pregnancy, and contraceptive use. To investigate age as a variable, a comparison was made between juveniles (0.5–1.5 years) and adults (>1.5–10 years). The age of 1.5 years was selected as a threshold, since marmosets are described to reach sexual maturity at this age [14]. Housing conditions include permanent indoor housing compared to housing with outdoor access. The effect of pregnancy status on blood values was investigated by comparing pregnant versus non-pregnant adult females. During the physical examinations the veterinarian determined if a marmoset was pregnant by means of abdominal palpation. There is a possibility that early gestation were missed using this method. In order to ensure that all pregnancies were included in the dataset, the gestation time of 144 days [14] was subtracted from the parturition date. The effect of contraception was determined by comparing contracepted females with non-pregnant adult females.

### 2.5. Statistics

A reference interval for a laboratory parameter can be defined by the limits of the parameter that includes 95% of laboratory values in a healthy population. The assumption is made that these 95% are physiological. The remaining 5% are therefore considered abnormal and include both 2.5% of the values below the lower limit of the reference interval and the 2.5% values above the upper limit of the reference interval. However, many of the blood parameters are not normally distributed, therefore reference interval calculations using the 2.5th and 97.5th percentiles using parametric methods (i.e., arithmetic mean ± 1.96 × standard deviations) are not correct. In this study, the reference intervals were determined by utilizing nonparametric methods to calculate the 2.5th and 97.5th percentiles for this reason. In addition, reference values were calculated using parametric methods. To this end, Box–Cox transformations were applied to obtain a normal distribution, and reference intervals were calculated using parametric methods. The final reference lower interval was established as the minimum value of the percentile or Box–Cox-transformed value and upper interval as the maximum of these values. In order to compare the size of the differences when comparing investigated variables, the percentual difference of the medians between the variables, hereafter referred to as Delta or Δ, were calculated as follows:$$100\times \frac{Median\;Group\;1\;–\;Median\;Group\;2}{Mean\;of\;the\;Median\;of\;Group\;1\;and\;the\;Median\;of\;Group\;2}$$

The statistical significance of between-group differences was evaluated using nonparametric tests (Mann–Whitney). *p*-values were corrected for multiple testing for the age-sex comparisons using Holm’s correction, and *p*-values ≤ 0.001 were considered statistically significant.

Age plots were determined for the HER and CER parameters. The correlation coefficient (*r*) was calculated, and the results were categorized as no correlation (*r* = 0), weak correlation (*r* < |0.3|), moderate correlation (|0.3| > *r* < |0.7|), and strong correlation (*r* > |0.7|).

## 3. Results

Hematological and biochemical reference intervals are presented in Table 3.

### 3.1. Influences on the Biochemical and Hematological Values

Data analysis revealed many statistically significant differences (delta) between the compared variables. However, in order to present the reader with a usable set of reference intervals, the decision was made to focus only on parameters that were judged clinically relevant. Delta values > 5% [15] and with a *p*-value ≤ 0.001 were regarded to be clinically relevant. Figures show all delta values calculated for the comparisons for each variable with the differences > 5% in bold.

#### 3.1.1. Effect of Age and Sex

Total bilirubin showed to be increased in juvenile males, compared to juvenile females (Δ = 8.1) (Figure 1). ALP (Δ = 25 in females, Δ = 24.6 in males), phosphate (Δ = 11.4 in females, Δ = 10.6 in males), white blood cell count (Δ = 6.7 in females, Δ = 5.6 in males), lymphocyte count (Δ = 10.1 in females, Δ = 8.2 in males), and basophil count (Δ = 16.7 for both sexes) were increased in both sexes for juveniles compared to adults. AST was increased in juvenile females compared to adult females (Δ = 5.3). GGT (Δ = 15.2 in females, Δ = 13.8 in males) and Fe (Δ = 6.2 in females, Δ = 8.5 in males) showed decreased values for both sexes in juveniles compared to adults.

Age plots were calculated for all the HER and CER parameters. None of the correlation coefficients were strong, although a moderate negative correlation was observed for ALP (rho = −0.587 for females and rho = −0.648 for males) and phosphate (rho = −0.51 for females and rho = −0.523 for males). The age–sex distribution of the biochemical parameters of bone turnover (calcium, phosphate, bicarbonate, and ALP) are depicted in detail in Figure 2A–D to emphasize the observed age effect related to bone metabolism.

#### 3.1.2. Effect of Housing Condition

For both sexes, increased levels of ALT (Δ = 14.5 in females, Δ = 12.7 in males) and GGT (Δ = 18.9 in females, Δ = 11.9 in males), but decreased levels of neutrophil count (Δ = 6.5 in females, Δ = 11.4 in males) and total bilirubin (Δ = 7.1 in females, Δ = 8.8 in males) were observed for marmosets housed strictly indoors compared to marmosets housed with outdoor access. LDH values were increased in females housed strictly indoors (Δ = 6.1) compared to females housed with outdoor access. Glucose (Δ = 5.2), WBC (Δ = 5.5) and monocyte count (Δ = 5) were decreased in males housed strictly indoors compared to males housed with outdoor access (Figure 3).

#### 3.1.3. Effect of Pregnancy and Contraception

Pregnant females showed increased ALP (Δ = 5.6), total bilirubin (Δ = 11.1), neutrophil count (Δ = 8.1), monocyte count (Δ = 10.7), and basophil count (Δ = 16.7) compared to non-pregnant females. Pregnant females also showed decreased ALT (Δ = 16.5), AST (Δ = 5.4), cholesterol (Δ = 7.5), Fe (Δ = 6.7), and lymphocyte count levels (Δ = 7) compared to non-pregnant females (Figure 4).

Eighty-one samples were collected from females contracepted with etonogestrel. Those females showed decreased P-LCR levels (Δ = 5.3) compared to adult non-pregnant females without an etonogestrel implant (Figure 5).

## 4. Discussion

Hematological and serum biochemistry values are important to monitor the health status of marmosets. Incorrect reference intervals act as a confounding variable. This affects the interpretation of results, potentially skewing the direction of future studies, resulting in a waste of resources and potential irreproducibility. The research reported here revealed that juveniles showed increased ALP, phosphate, WBC, lymphocyte count, and basophil count and decreased levels of GGT and Fe compared to adults. Marmosets housed strictly indoors showed increased ALT and GGT and decreased levels of total bilirubin and neutrophil count compared to marmosets housed with outdoor access. Pregnant marmosets showed increased ALP, total bilirubin, neutrophil count, monocyte count, and basophil count, and decreased levels of AST, ALT, cholesterol, Fe, and lymphocyte count compared to non-pregnant marmosets. Etonogestrel-contracepted females showed decreased P-LCR values compared to adult non-pregnant females who were not contracepted. Knowing the reference intervals and the impact of sex, age, housing, contraceptive use, and pregnancy status on these intervals aids researchers and veterinarians in making well-informed decisions.

Previously, different studies have reported reference values for HER and CER for common marmosets [2,3,4,5,6,7,8]. However, no report analyzes all parameters and compares all variables we included and used a large sample size while making use of the appropriate statistical methods. It is important to consider calculating reference values for each individual facility, using their own animals, diet, sedation protocols, catching methods, etc., as these values can influence laboratory parameters [12,13,16,17,18]. However, in this dataset it was attempted to exclude and investigate the effect of as many variables as possible to achieve a representative and statistically correct collection of reference intervals to be used at facilities that do not have the means to calculate them themselves.

Total bilirubin showed to be increased in juvenile males compared to juvenile females. This has also been described by other research in marmosets [2]. A similar gender effect has been observed in humans associated with fasting, although the underlying mechanism of fasting associated hyperbilirubinemia remains unknown. Further investigation of the relationship between biochemical analytes and fasting is required [18]. In humans, HGB and HCT concentrations are lower in women compared to men [19]. This difference was not observed in marmosets, likely because marmosets do not menstruate, nor are there any external signs of ovulation.

For both sexes, increased levels of ALP, phosphate, WBC, lymphocyte count, and basophil count were observed for juveniles compared to adults. Increased levels of ALP in juvenile animals compared to adults have been described across species, including marmosets [2,6] and other nonhuman primates [15], and is the result of increased bone iso-enzyme activity associated with growth [20,21,22]. ALP and phosphate are markers of bone metabolism. Hence, growing animals show physiologically increased levels of these markers. The age curves for ALP and phosphate in Figure 2A,D show a significant moderate negative trend. A steep downward trend can be observed from the age of 0.5 until approximately 2.5 years, at which point the curve reaches a plateau. This contrasts with the cut-off age of 1.5 years commonly used when dividing the juvenile from the adult marmosets based on sexual maturity [14]. However, during the first year of sexual maturity, marmosets still show a significant difference for ALP and phosphate. Although no relevant differences (delta < 5%) in other bone turnover markers were observed, i.e., calcium and bicarbonate, a small positive age correlation for bicarbonate was observed. This is depicted in Figure 2C. Other bone metabolism markers, such as PTH and vitamin D3 were not included in this analysis. The increased levels of WBC, lymphocyte count, and basophil count observed in juveniles may be attributed to an increased exposure to foreign antigens during childhood [23]. At this facility, all juvenile animals are housed with outdoor access, which can be a possible confounder for this observed effect, since results showed that marmosets housed with outdoor access also had higher neutrophile levels in both sexes and males also showed higher levels of WBC and monocytes. Similar findings are described in the literature about marmosets as well [2]. Other research in marmosets showed higher levels of monocyte count and eosinophile count and lower levels of glucose and RBC in juvenile males compared to adult males, higher levels of Ca, HGB, and RBC in juvenile females compared to adult females [2,5]. Adult marmosets of both sexes showed increased levels of GGT and Fe compared to juveniles. Similarly, GGT levels have been demonstrated to increase with age in humans [24]. The increase of Fe level with age contrasts with observations in humans, where an inverse correlation is described [25]. In marmosets, higher levels of Fe in adult males compared to juvenile males have been previously described [6]. In rhesus and cynomolgus macaques, a similar effect has also been observed [15]. Other research groups have also reported higher levels of cholesterol, creatinine, red blood cells, HGB, HCT [2,7] in males compared to females and higher levels of MCV, MCH [7], and PLT [5], which we did not conclude in this article. Possible explanations to this difference could be in the different statistical methods used, sample size, sedation protocols, or other variables that were different between institutions.

Geriatric animals were not included as a variable when calculating reference intervals split by age. There is no clear definition of at what age a marmoset becomes geriatric. The average and maximum life spans also differ greatly among institutions. Research at various institutions show an average lifespan ranging from 5 to 7 years to 10 to 12 years [26,27]. In Appendix A, age is depicted as a continuous scale to provide some additional information for those interested in the relation between laboratory parameters and aging in marmosets. However, it should be noted that the sample size above 10 years of age is relatively small in this dataset.

Marmosets housed with outdoor access showed increased total bilirubin and neutrophil count compared to strictly indoor housed marmosets, while strictly indoor housed marmosets showed increased ALT and GGT levels compared to marmosets housed with outdoor access. However, the influence of pregnancy on the observed data (see Section 3.1.3) must be considered, as pregnant females were not excluded from the indoor–outdoor housed marmosets in this comparison.

Based on literature, a period of 6 weeks to 3 months is required for NHPs to adjust to new conditions after transport, including the normalization of the HER and CER parameters [28,29,30,31,32]. However, the number of blood samples collected in this study within three months following relocation within the BPRC facility from housing with outdoor access to strictly indoor housing was relatively small in comparison to the total number of blood samples, reducing the effect of this potential confounder.

Increased levels of ALP, total bilirubin, neutrophil count, monocyte count, and basophil count, and decreased levels of AST, ALT, cholesterol, Fe and lymphocyte count were observed in pregnant females compared to non-pregnant females. These alterations have also been observed in humans, with the exception of the total bilirubin and cholesterol, which showed an inverse relation with pregnancy [33]. In marmosets an increase of neutrophil count and a decrease in ALT have previously also been observed. However, they also reported a decrease of bicarbonate, urea, and total bilirubin, which contrasts with our findings [2]. The reason for this difference cannot be explained and is most likely attributed to a difference in facilities and housing. In rhesus macaques, a similar effect of lymphocyte count and ALT, as well as a similar basophil count, monocyte count, and neutrophil count effect in cynomolgus macaques [15].

The research reported here was the first to test the effect of contraception on blood-based parameters in common marmosets. In females contracepted with etonogestrel, a decrease in P-LCR compared to non-contracepted females was observed. In humans, a significant difference in ALT, urea, PCV and ALP has been described with the use of etonogestrel as a contraceptive [34]. In rhesus macaques, the effect of an etonogestrel implant was investigated and no effect of the implant was found on the blood chemistry parameters [35].

The method of sedation was uniform for all samples included in order to minimize external influences other than the investigated variables. Alphaxalone has preference over ketamine and ketamine–medetomidine sedation protocols in marmosets due to the better depth of the anesthesia and a smoother recovery [12,13]. The use of alphaxalone could, however, have a possible effect on the laboratory values, but the literature investigating this in this species is scarce. In cats, the effect of alphaxalone on biochemical and blood–gas values has been investigated and showed no significant changes after sedation [36]. Muscle damage indicators (CK, AST, and LDH) for repeated sedations with alphaxalone were increased the least compared to repeated sedation with ketamine or ketamine–medetomidine in marmosets [12]. In rhesus macaques, the addition of medetomidine to ketamine causes an increase in glucose levels compared to sedation with only ketamine [15]. This effect has not been described for alphaxalone. The data used in this article have been collected using a single sedation, therefore it is not expected that muscle damage indicators would be increased because of the sedation. Researchers and veterinarians working with these sedation protocols should keep this in mind when interpreting the reference values calculated in this article. It remains important for the reproducibility of research to include as much information as is necessary about the manner in which the animals are kept and handled, especially including sedation protocols.

Body condition score was not included as a variable. Over the years, different veterinarians evaluated the marmosets during blood sampling, which may introduce assessor bias. BCS was only used to exclude marmosets that were severely obese (BCS > 4.5) and marmosets that were heavily underweight (BCS < 1.5), as these conditions are known to affect blood parameters [37,38]. For these marmosets, underlying comorbidities could not be ruled out.

## 5. Conclusions

Information on the effect of age, sex, housing conditions, contraceptive use, and pregnancy status on laboratory parameters in marmosets will aid researchers and veterinarians in identifying physiological and pathological changes.

The availability of updated reference intervals, calculated using a large sample size and using the appropriate statistical methods, is essential to provide optimum care to patients and to support biomedical research in marmosets.

## Figures and Tables

**Figure 1 animals-14-00790-f001:**
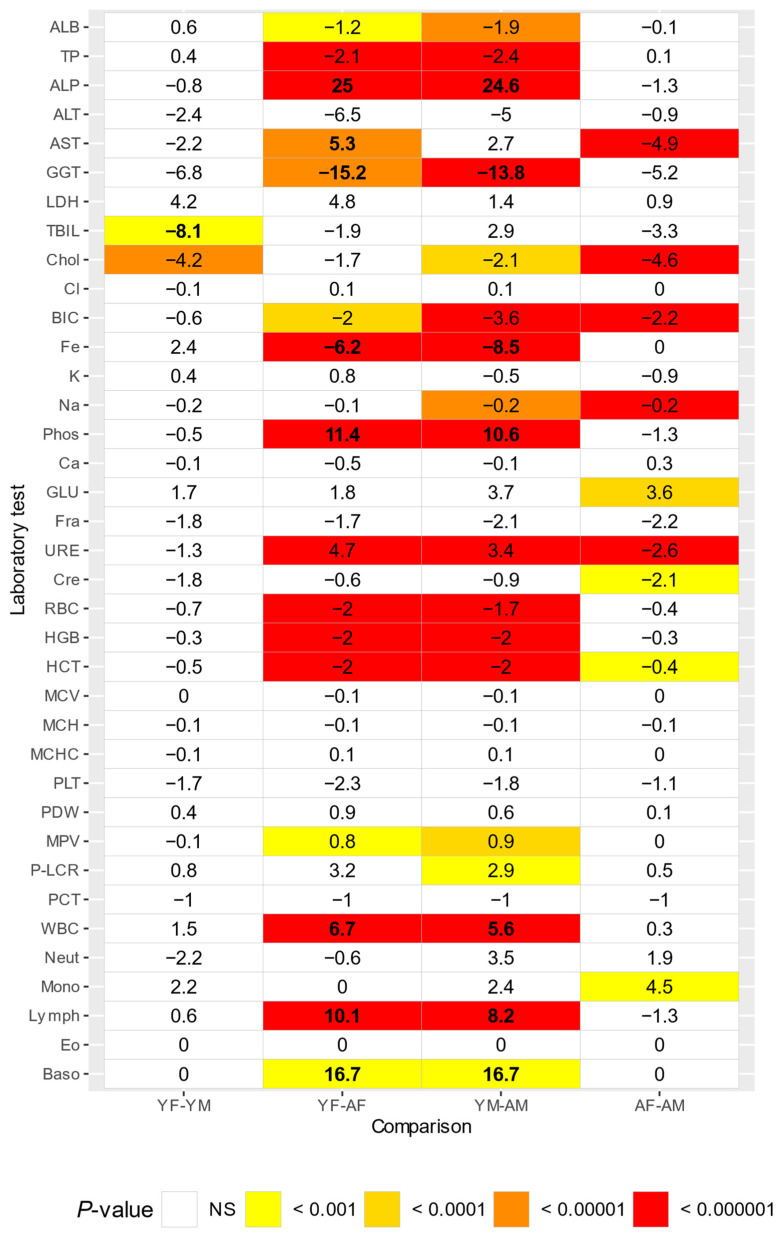
Comparisons between sex and age for marmosets, numbers indicate percentual differences (Delta) between groups. Delta values greater than 5% are in bold. YF-YM = young females and males, YF-AF = young and adult females, YM-AM = young and adult males, AF-AM = adult females and males. NS = not significant.

**Figure 2 animals-14-00790-f002:**
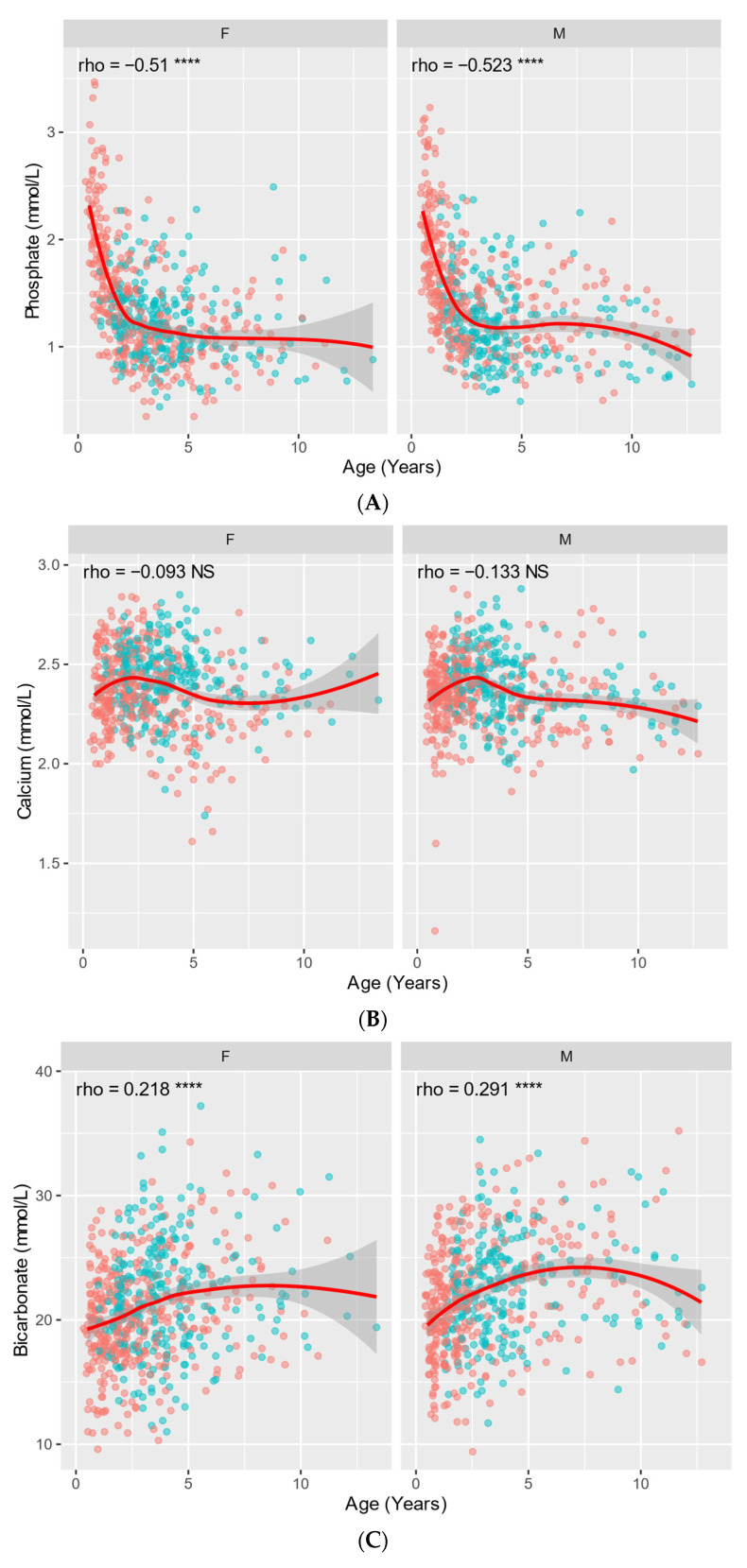
Age plot of phosphate (**A**), calcium (**B**), bicarbonate (**C**), and alkaline phosphatase (**D**) between sex and age for marmosets. F = female, M = male. Points represent individual animals (red = outdoor access, cyan = indoor only). The red line indicates a Loess moving average and the grey area indicates the 95% confidence interval. Rho is Spearman’s rank correlation for parameter and Age; NS = not significant, asterisks indicate statistical significance levels **** < 0.000001.

**Figure 3 animals-14-00790-f003:**
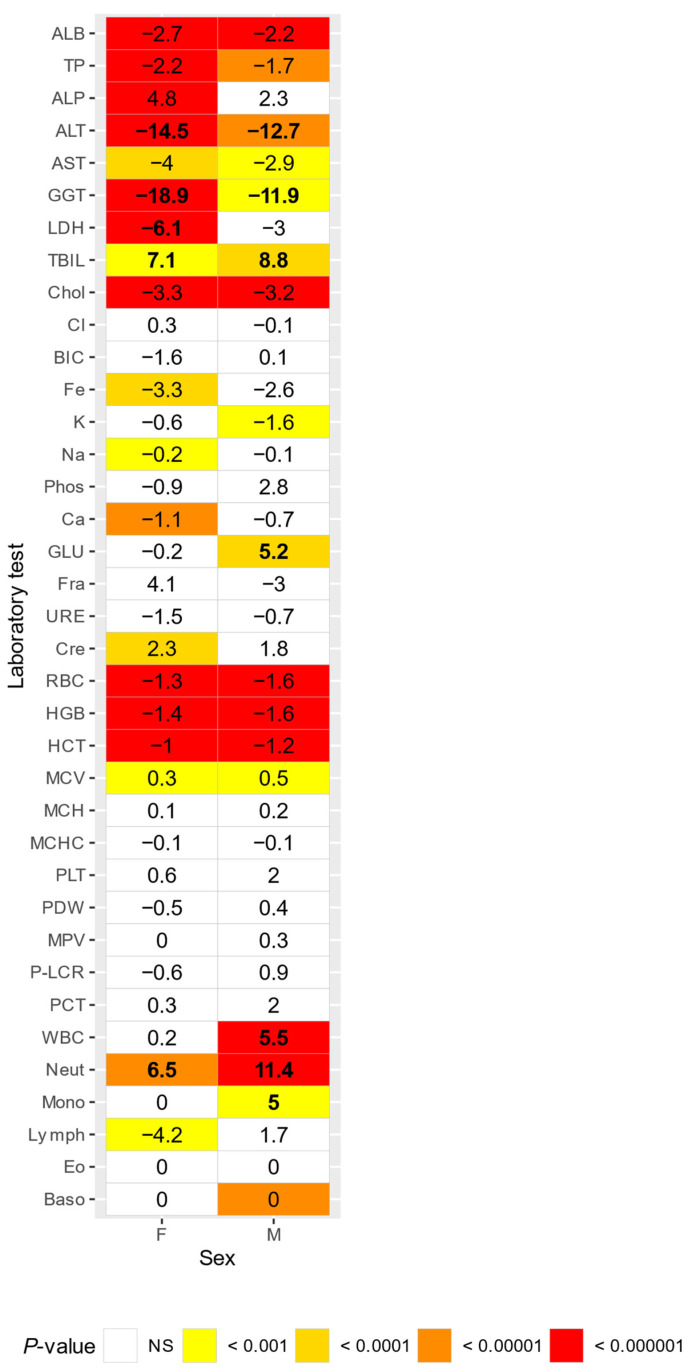
Comparison of adult marmosets with outdoor access or housed strictly indoors. Numbers indicate percentual differences (Delta) between groups. Delta values greater than 5% are in bold. NS = not significant.

**Figure 4 animals-14-00790-f004:**
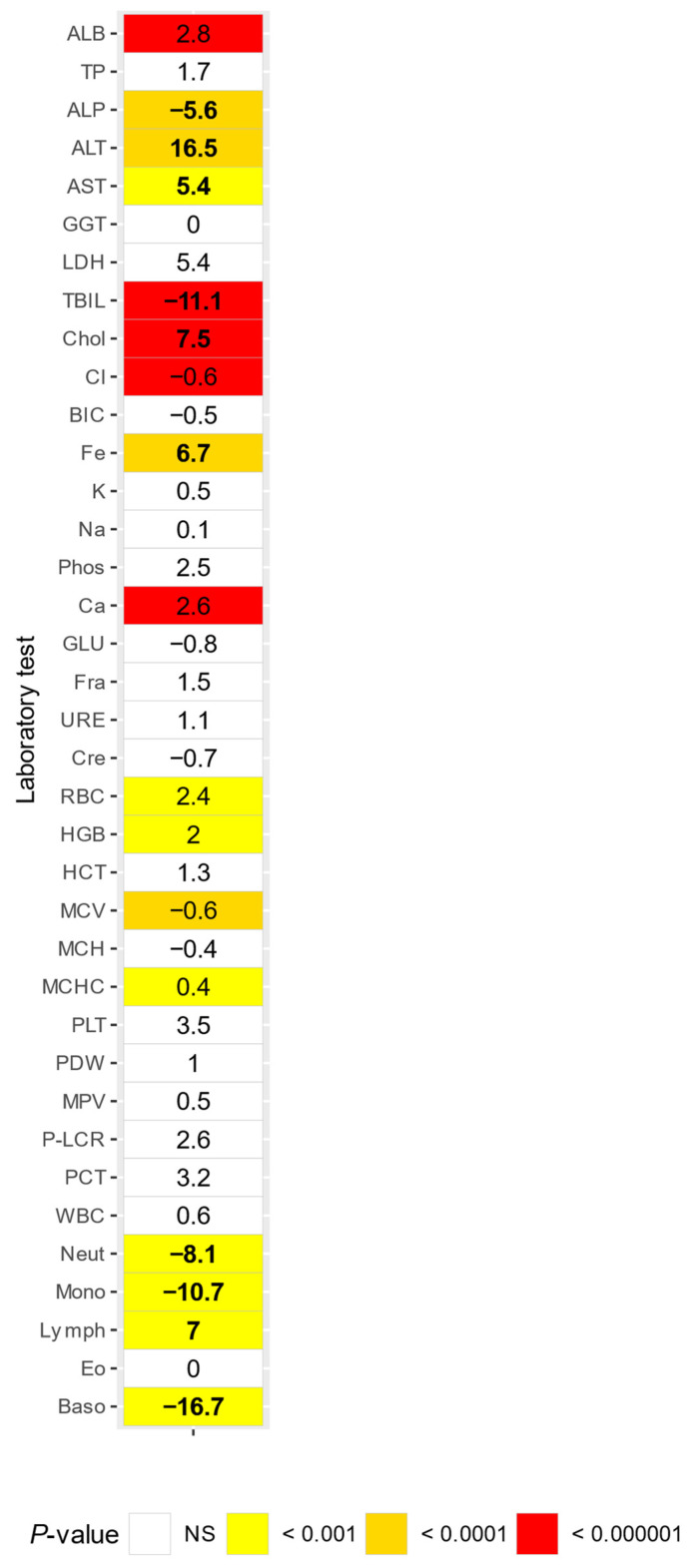
Comparison of non-pregnant or pregnant adult females, numbers indicate percentual differences (Delta) between, Delta values greater than 5% are in bold. NS = not significant.

**Figure 5 animals-14-00790-f005:**
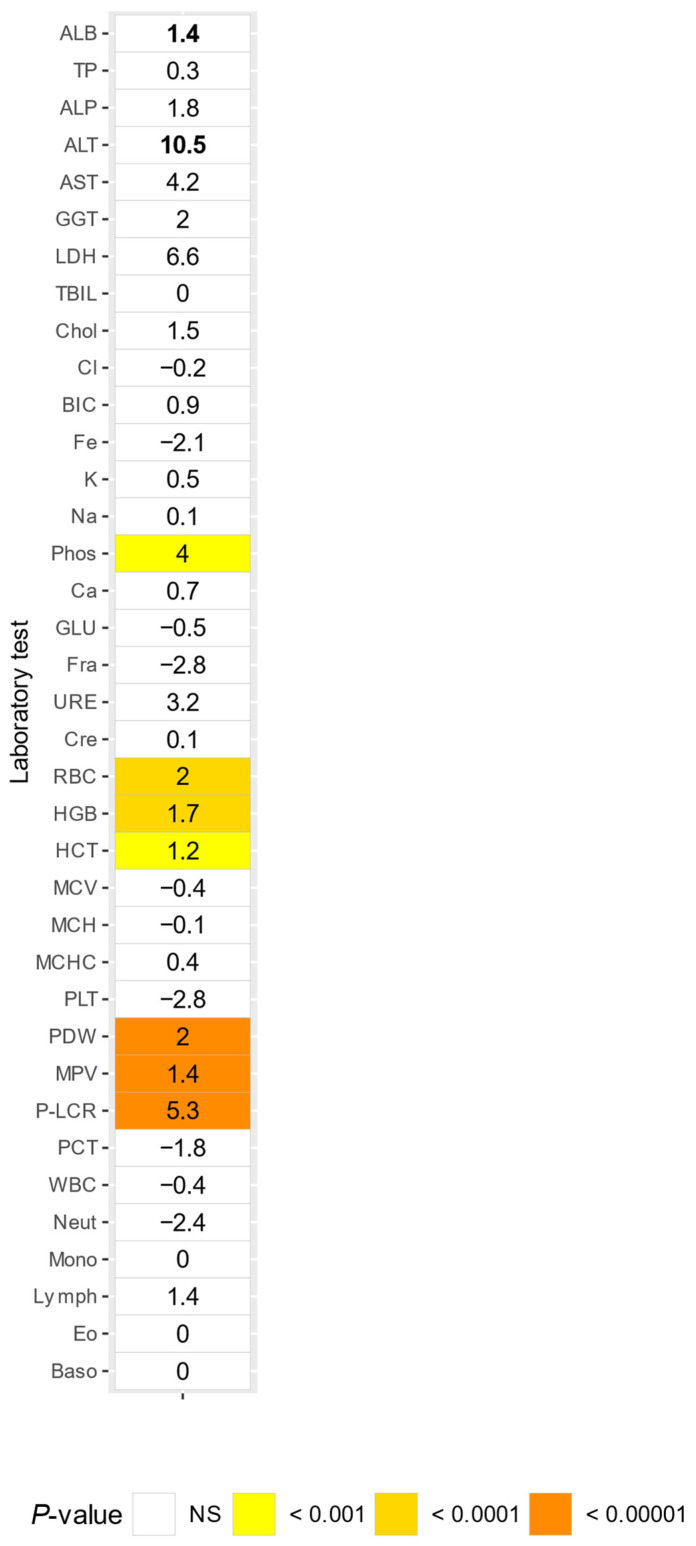
Comparison of adult females without or with an etonogestrel contraceptive, numbers indicate percentual differences (Delta) between, Delta values greater than 5% are in bold. NS = not significant.

**Table 1 animals-14-00790-t001:** Distribution of age and sex of the unique animals in the dataset.

	Juvenile (<1.5 years)	Adult (≥1.5 years)	Total
Female	127	132	259
Male	138	75	213
Total	265	207	472

**Table 2 animals-14-00790-t002:** Number of repeats per animal and the distribution by sex.

Repeats	Female	Male	Total	Nr of Observations
1	83	46	129	129
2	76	61	137	274
3	36	41	77	231
4	31	28	59	236
5	18	13	31	155
6	2	6	8	48
7	6	8	14	98
8	7	7	14	112
9	0	3	3	27
Total	259	213	472	1310

**Table 3 animals-14-00790-t003:** Hematological and serum biochemical reference intervals and percentiles in the common marmoset. P2.5 refers to the 2.5th percentile. P97.5 refers to the 97.5th percentile. RI low refers to lower reference interval, and RI upp refers to upper reference interval.

Parameter	Sex	Age	Count	P2.5	P97.5	Box Cox Low	Box Cox Upp	RI Low	RI Upp
ALB	F	<1.5	128	29.6	48.6	30.0	48.9	29.6	48.9
(g/L)	F	≥1.5	509	29.1	53.0	28.1	53.6	28.1	53.6
	M	<1.5	140	30.7	46.6	30.8	47.5	30.7	47.5
	M	≥1.5	439	29.3	53.0	28.8	53.7	28.8	53.7
TP	F	<1.5	128	41.3	64.4	41.6	65.2	41.3	65.2
(g/L)	F	≥1.5	513	39.8	68.1	40.8	68.7	39.8	68.7
	M	<1.5	147	41.1	62.4	40.9	62.0	40.9	62.4
	M	≥1.5	444	40.2	68.1	40.3	68.4	40.2	68.4
ALP	F	<1.5	128	89.1	526.2	91.1	554.5	89.1	554.5
(U/L)	F	≥1.5	512	41.8	206.8	40.4	201.1	40.4	206.8
	M	<1.5	147	102.5	654.0	89.5	712.6	89.5	712.6
	M	≥1.5	445	49.1	172.4	48.5	176.4	48.5	176.4
ALT	F	<1.5	128	0.60	75.27	0.41	80.32	0.41	80.32
(U/L)	F	≥1.5	512	1.30	175.76	1.01	159.39	1.01	175.76
	M	<1.5	146	0.78	125.50	0.62	115.02	0.62	125.50
	M	≥1.5	445	1.22	149.78	1.15	140.98	1.15	149.78
AST	F	<1.5	128	98.0	453.0	92.0	523.8	92.0	523.8
(U/L)	F	≥1.5	512	68.9	591.1	69.9	510.6	68.9	591.1
	M	<1.5	147	96.6	819.0	95.7	741.9	95.7	819.0
	M	≥1.5	445	80.1	457.5	80.9	495.4	80.1	495.4
GGT	F	<1.5	128	0.0	12.0	0.0	11.6	0.0	12.0
(U/L)	F	≥1.5	513	0.0	28.4	0.0	23.7	0.0	28.4
	M	<1.5	147	0.0	7.6	0.0	10.4	0.0	10.4
	M	≥1.5	445	0.0	17.2	0.0	19.3	0.0	19.3
LDH	F	<1.5	128	160.9	2312.5	154.0	2558.8	154.0	2558.8
(U/L)	F	≥1.5	512	139.3	2681.4	136.8	2492.5	136.8	2681.4
	M	<1.5	147	150.9	3092.8	139.2	2689.2	139.2	3092.8
	M	≥1.5	445	130.1	1572.6	125.9	1693.1	125.9	1693.1
TBIL	F	<1.5	128	0.00	1.88	0.01	2.13	0.00	2.13
(mmol/L)	F	≥1.5	513	0.00	2.42	0.02	2.46	0.00	2.46
	M	<1.5	147	0.00	2.34	0.08	2.33	0.00	2.34
	M	≥1.5	445	0.00	2.10	0.04	2.25	0.00	2.25
CHOL	F	<1.5	128	2.07	4.36	1.98	4.55	1.98	4.55
(mmol/L)	F	≥1.5	512	1.75	5.01	1.69	5.00	1.69	5.01
	M	<1.5	147	2.38	5.28	2.40	5.27	2.38	5.28
	M	≥1.5	445	2.46	5.42	2.39	5.52	2.39	5.52
Cl	F	<1.5	126	100.0	113.1	101.1	114.1	100.0	114.1
(mmol/L)	F	≥1.5	512	100.4	114.0	100.8	114.3	100.4	114.3
	M	<1.5	144	100.7	113.1	102.0	114.2	100.7	114.2
	M	≥1.5	443	102.0	113.9	102.2	113.5	102.0	113.9
BIC	F	<1.5	128	11.2	26.9	11.6	26.7	11.2	26.9
(mmol/L)	F	≥1.5	513	13.2	30.4	13.4	30.6	13.2	30.6
	M	<1.5	147	13.1	28.0	13.3	27.9	13.1	28.0
	M	≥1.5	445	14.4	31.9	14.5	31.4	14.4	31.9
Fe	F	<1.5	128	8.0	33.0	7.6	34.1	7.6	34.1
(mmol/L)	F	≥1.5	513	9.7	45.1	10.1	47.4	9.7	47.4
	M	<1.5	147	8.0	32.8	7.5	33.7	7.5	33.7
	M	≥1.5	444	10.9	44.3	10.0	46.1	10.0	46.1
K	F	<1.5	126	2.37	5.02	2.20	4.97	2.20	5.02
(mmol/L)	F	≥1.5	510	2.28	4.50	2.29	4.34	2.28	4.50
	M	<1.5	142	2.31	4.80	2.25	4.71	2.25	4.80
	M	≥1.5	442	2.29	4.59	2.30	4.47	2.29	4.59
Na	F	<1.5	126	139.5	151.6	138.7	154.6	138.7	154.6
(mmol/L)	F	≥1.5	512	141.0	153.9	141.5	154.4	141.0	154.4
	M	<1.5	144	139.9	152.7	140.6	154.7	139.9	154.7
	M	≥1.5	443	141.7	154.1	143.4	154.9	141.7	154.9
Phos	F	<1.5	128	0.97	3.04	1.03	3.11	0.97	3.11
(mmol/L)	F	≥1.5	513	0.61	2.03	0.60	2.03	0.60	2.03
	M	<1.5	147	1.14	3.02	1.08	3.10	1.08	3.10
	M	≥1.5	445	0.69	2.07	0.65	2.11	0.65	2.11
Ca	F	<1.5	128	2.04	2.69	2.03	2.73	2.03	2.73
(mmol/L)	F	≥1.5	513	1.97	2.75	1.99	2.75	1.97	2.75
	M	<1.5	147	1.98	2.65	1.97	2.68	1.97	2.68
	M	≥1.5	445	2.03	2.73	2.04	2.74	2.03	2.74
Glu	F	<1.5	128	2.6	18.2	2.6	19.6	2.6	19.6
(mmol/L)	F	≥1.5	513	2.3	15.3	2.1	15.6	2.1	15.6
	M	<1.5	147	2.2	17.1	2.1	19.6	2.1	19.6
	M	≥1.5	444	2.1	13.8	2.1	14.1	2.1	14.1
URE	F	<1.5	128	4.04	11.29	4.12	11.36	4.04	11.36
(mmol/L)	F	≥1.5	512	2.87	11.07	2.87	10.94	2.87	11.07
	M	<1.5	147	4.62	12.94	4.63	12.57	4.62	12.94
	M	≥1.5	444	3.22	11.65	3.42	12.02	3.22	12.02
CRE	F	<1.5	127	12.33	35.73	11.45	37.48	11.45	37.48
(mmol/L)	F	≥1.5	504	11.76	36.71	11.79	36.50	11.76	36.71
	M	<1.5	147	12.69	38.24	12.83	38.45	12.69	38.45
	M	≥1.5	437	12.59	39.43	12.61	39.01	12.59	39.43
RBC	F	<1.5	116	4.78	6.81	4.67	6.87	4.67	6.87
(10^12^/L)	F	≥1.5	496	4.75	7.40	4.86	7.48	4.75	7.48
	M	<1.5	134	4.57	7.03	4.67	7.06	4.57	7.06
	M	≥1.5	463	4.86	7.58	4.88	7.66	4.86	7.66
HGB	F	<1.5	116	6.95	9.51	6.84	9.78	6.84	9.78
(mmol/L)	F	≥1.5	496	7.00	10.56	7.21	10.65	7.00	10.65
	M	<1.5	134	6.60	9.94	7.02	10.00	6.60	10.00
	M	≥1.5	463	7.10	10.65	7.39	10.82	7.10	10.82
HCT	F	<1.5	116	0.347	0.460	0.342	0.467	0.342	0.467
(L/L)	F	≥1.5	496	0.357	0.505	0.364	0.510	0.357	0.510
	M	<1.5	134	0.324	0.481	0.343	0.483	0.324	0.483
	M	≥1.5	463	0.361	0.509	0.370	0.519	0.361	0.519
MCV	F	<1.5	116	63.7	76.7	63.7	77.6	63.7	77.6
(fL)	F	≥1.5	496	64.5	78.2	64.1	78.5	64.1	78.5
	M	<1.5	134	64.4	78.1	63.9	78.3	63.9	78.3
	M	≥1.5	463	64.1	78.8	64.1	78.6	64.1	78.8
MCH	F	<1.5	116	1324	1558	1303	1564	1303	1564
(amol)	F	≥1.5	496	1323	1576	1312	1571	1312	1576
	M	<1.5	134	1339	1572	1334	1573	1334	1573
	M	≥1.5	463	1321	1566	1317	1573	1317	1573
MCHC	F	<1.5	116	19.6	21.6	19.1	22.0	19.1	22.0
(mmol/L)	F	≥1.5	496	19.1	21.6	19.2	21.8	19.1	21.8
	M	<1.5	134	19.5	21.4	19.5	21.7	19.5	21.7
	M	≥1.5	463	19.2	21.5	19.2	21.8	19.2	21.8
RDW-SD	F	<1.5	116	32.4	40.7	31.8	39.8	31.8	40.7
(fL)	F	≥1.5	496	32.3	43.7	31.7	42.0	31.7	43.7
	M	<1.5	134	32.8	41.7	32.3	41.1	32.3	41.7
	M	≥1.5	463	32.3	43.3	31.8	41.4	31.8	43.3
RDW-CV	F	<1.5	116	12.9	17.8	12.6	18.5	12.6	18.5
(%)	F	≥1.5	496	12.7	19.1	12.6	19.7	12.6	19.7
	M	<1.5	134	12.9	18.1	12.9	18.8	12.9	18.8
	M	≥1.5	463	12.7	18.8	12.4	19.8	12.4	19.8
PLT	F	<1.5	116	95.5	893.5	112.4	880.3	95.5	893.5
(10^9^/L)	F	≥1.5	496	172.1	831.4	195.7	827.9	172.1	831.4
	M	<1.5	134	105.9	818.8	141.7	802.3	105.9	818.8
	M	≥1.5	463	131.0	862.0	193.8	884.2	131.0	884.2
PDW	F	<1.5	112	8.4	14.4	8.2	14.5	8.2	14.5
(fL)	F	≥1.5	495	8.4	13.2	8.4	13.4	8.4	13.4
	M	<1.5	130	8.7	15.1	8.5	14.0	8.5	15.1
	M	≥1.5	458	8.6	13.2	8.5	13.2	8.5	13.2
MPV	F	<1.5	112	8.2	11.2	8.2	11.6	8.2	11.6
(fL)	F	≥1.5	495	8.1	10.8	8.1	10.9	8.1	10.9
	M	<1.5	130	8.4	11.4	8.3	11.4	8.3	11.4
	M	≥1.5	458	8.1	10.8	8.1	10.8	8.1	10.8
P-LCR	F	<1.5	112	11.5	34.9	11.0	37.8	11.0	37.8
(%)	F	≥1.5	495	10.7	32.5	10.7	32.9	10.7	32.9
	M	<1.5	130	13.0	36.7	12.5	35.8	12.5	36.7
	M	≥1.5	458	11.1	31.8	10.9	32.5	10.9	32.5
PCT	F	<1.5	112	0.193	0.832	0.177	0.799	0.177	0.832
(%)	F	≥1.5	495	0.181	0.750	0.206	0.746	0.181	0.750
	M	<1.5	130	0.180	0.730	0.203	0.739	0.180	0.739
	M	≥1.5	458	0.194	0.800	0.217	0.793	0.194	0.800
WBC	F	<1.5	116	2.22	11.66	2.19	11.24	2.19	11.66
(10^9^/L)	F	≥1.5	495	1.57	8.57	1.68	8.27	1.57	8.57
	M	<1.5	134	2.23	10.61	2.14	9.97	2.14	10.61
	M	≥1.5	463	1.47	7.72	1.46	7.76	1.46	7.76
Neut	F	<1.5	109	0.76	5.14	0.73	5.61	0.73	5.61
(10^9^/L)	F	≥1.5	476	0.55	5.75	0.57	5.66	0.55	5.75
	M	<1.5	131	0.65	5.90	0.64	5.73	0.64	5.90
	M	≥1.5	442	0.49	5.38	0.49	5.32	0.49	5.38
Mono	F	<1.5	109	0.040	0.516	0.042	0.457	0.040	0.516
(10^9^/L)	F	≥1.5	477	0.030	0.431	0.026	0.405	0.026	0.431
	M	<1.5	131	0.020	0.370	0.023	0.349	0.020	0.370
	M	≥1.5	443	0.020	0.310	0.023	0.304	0.020	0.310
Lymph	F	<1.5	109	1.08	7.48	0.97	7.18	0.97	7.48
(10^9^/L)	F	≥1.5	477	0.55	4.13	0.47	4.03	0.47	4.13
	M	<1.5	131	0.99	5.90	1.03	5.55	0.99	5.90
	M	≥1.5	443	0.59	3.95	0.58	3.95	0.58	3.95
Eo	F	<1.5	109	0.000	0.176	0.001		0.000	0.176
(10^9^/L)	F	≥1.5	439	0.000	0.121	0.001		0.000	0.121
	M	<1.5	131	0.000	0.118	0.001		0.000	0.118
	M	≥1.5	427	0.000	0.154	0.001		0.000	0.154
Baso	F	<1.5	116	0.000	0.060	0.001	0.061	0.000	0.061
(10^9^/L)	F	≥1.5	487	0.000	0.058	0.001	0.054	0.000	0.058
	M	<1.5	134	0.000	0.090	0.001	0.084	0.000	0.090
	M	≥1.5	458	0.000	0.050	0.001	0.055	0.000	0.055

## Data Availability

Data are available upon reasonable request.

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
