# Peer review of "Hematological and Serum Biochemical Reference Intervals for Alphaxalone Sedated Common Marmosets (Callithrix jacchus)"

_animals, 2024, doi:10.3390/ani14050790_

Round 1

Reviewer 1 Report

Comments and Suggestions for Authors

This manuscript reports reference intervals for a wide variety of hematological and serum biomarkers in common marmosets. It is notable for the size of the sample, the known health of the individuals included, and the similarity in treatment across animals. The article is well-written and concise. My comments are minor.

Line 55 and throughout: As these are animals, “sex” should be used instead of “gender.”

Section 2.3: What is the typical lifespan of captive marmosets? Is there a benefit to subdividing adults so that the authors can compare geriatric individuals compared to those in their prime? Or in looking at age as a continuous rather than categorical variable?

Figures 1, 3, 4 and 5: This is an interesting way to present these differences but I think most readers would find tables with the averages, deltas, and then statistical results to be more informative. The p-values in the statistical results section could still be color-coded if the author’s desire to visualize the magnitude of difference.

Reviewer 2 Report

Comments and Suggestions for Authors

See attached file.

Comments on the Quality of English Language

Aside from a few grammatical errors, the quality of the English in this manuscript is superb.

Reviewer 3 Report

Comments and Suggestions for Authors

Summary

This manuscript establishes reference intervals for hematologic and serum biochemical parameters in marmosets anesthetized with alphaxalone. The RIs were established using data collected from 472 unique individuals, and sampling was repeated annually for a total of 1310 blood samples. Differences were observed with stratification by sex, age, housing, contraceptive use, and pregnancy, and authors provide biological explanations for these differences. This manuscript provides marmoset RIs based on the largest number of individuals to date and employs a consistent anesthetic protocol for all sample collections. Authors employed two methods to calculate the RIs and then used the widest resulting range of values.

General comments

The inclusion of repeated samples from some individuals allows for the undue influence of those individuals on the reference intervals. This may be an issue if any of the individuals have values that tend to be at the extremes of the population. While the investigators did exclude animals with overt signs of disease, clinicopathological evidence of disease often predates overt disease and minor changes to parameters may be seen in advance of many diseases such as chronic kidney disease. Please address the decision to include repeat samples on animals and the potential effects this could have on the intervals.

The methods section does not indicate the method used to identify and exclude outliers, which is an important step in the generation of reference interval guidelines and should be mentioned (see the ASVCP reference interval guidelines). Since the authors acknowledge that most laboratory values are not normally distributed, they initially calculated the RIs using nonparametric methods, but they need to include a statement indicating that they checked the distribution of each parameter before and after doing the Box-Cox transformation to confirm that data was not normally distributed prior to the transformation and was normal after the transformation. I am not clear on why the authors chose to calculate the reference intervals using a second method as nonparametric methods are recommended when more than 120 samples are available for each group (see ASVCP reference interval guidelines). Sample sizes were not included for all methods of stratification so if there are groups with smaller sample sizes please refer to those guidelines for the appropriate statistical test to use.

Table 3 includes the sample size for the groups stratified by sex and age, but the sample size also needs to be included for the later stratifications by housing condition, reproductive status of animals, and birth control use. Since the authors state that delta values > 5% and p values of 0.001 are considered clinically relevant, Figures 1 and 3 could be moved to the supplementary figures section and replaced with tables styled after Table 3 including the actual RI’s calculated for the stratifications of data by housing condition and reproductive and contraceptive status.

Specific Comments

Line 236 I am not familiar with using the percentual difference of the medians to compare differences between groups so please include a reference for deciding that >5% difference is clinically relevant.

Line 258 Figure 1 does not seem necessary as part of the main body of the paper and could be moved to the supplementary figures section.

Line 281 Please include the sample size for the housing condition data. Referring to the percentual differences between housing conditions is not meaningful information without including the actual reference intervals. I suggest including tables with the reference intervals here or in the supplementary figures.

Line 294 Figure 3 does not seem necessary as part of the main body of the paper and could be moved to the supplementary figures section in favor of a table including the actual reference intervals for these lab tests.

Line 299 Please include the sample sizes and the reference intervals for this data.

Line 311 Figure 4 could be moved to the supplementary figures section in favor of a table including the actual reference intervals for these lab tests.

Line 316 Figure 5 could be moved to the supplementary figures section in favor of a table including the actual reference intervals for these lab tests.

Lines 336-341 These sentences imply that there is no use creating and publishing reference intervals for readers to use since there are always differences in collection methodology between institutions. While I agree that there will likely be differences between institutions, it would be more valuable to the reader to compare the results in this paper to the previous publications referenced to see which findings are consistent across institutions and publications. It is appropriate to point out marked differences between the results of this paper and previous publications in the text of the discussion rather than excluding the findings of those papers entirely.
